# Enabling Low-Resource Transfer Learning across COVID-19 Corpora by Combining Event-Extraction and Co-Training

**Alexander Spangher, Nanyun Peng, Jonathan May, Emilio Ferrara**
Information Sciences Institute, University of Southern California, Marina del Rey, USA
{spangher, npeng, jonmay, ferrarae}@isi.edu

## Abstract

Social-science investigations can benefit from a direct comparison of heterogenous corpora: in this work, we compare U.S. state-level COVID-19 policy announcements with policy discussions on Twitter. To perform this task, we require classifiers with high transfer accuracy to both (1) classify policy announcements and (2) classify tweets. We find that co-training using event-extraction views improves the transfer accuracy of our RoBERTa classifier by 11% above a baseline. The same improvements are not observed with a baseline classifier, for the baseline classification task or for baseline views. With only a small set of 576 COVID-19 policy announcements, hand-classified into 1 of 6 categories, our RoBERTA co-trained classifier observes a maximum transfer accuracy of .77 f1-score on a hand-validated set of tweets. This work represents the first known application of these techniques to an NLP transfer learning task and facilitates cross-corpora comparisons necessary for studies of social science phenomena.

## 1 Introduction

During the initial stages of the COVID-19 crisis, the U.S. lacked a centralized political response and a cultural familiarity with pandemics. From a social science perspective, research into COVID-19 policies and conversations around policy has consequences for both citizens and policy-makers as: (1) the extent of citizens' adherence to policy is often determined by awareness, and (2) in a democracy, policy-makers aim to produce policy that citizens are likely to support.

*To lay the methodological groundwork for a such a comparative study of policy and conversation*, we examine methods for cross-corpora transfer learning across COVID-19 government policy statements and COVID-19 tweets. Such a task is

| State-Level Policy | Label |
|---|---|
| The Governor issued a "Stay Safe, Stay Home" Directive. | Public Space Restriction |
| The Governor announced [loans] to provide relief for restaurants, bars, businesses. | Economic Measures |
| The Indiana State Department of Health announced new testing for COVID-19. | Healthcare |
| **Tweets** | **Label** |
| Can You Cancel Your Flight Because of the Coronavirus? | Travel Restrictions |
| Everyone should start wearing face masks in crowded public places. | Healthcare |
| What will it take for restaurants and ALL non-essential businesses to close? | Public Space Restriction |

Table 1: Sample Policy Announcements (top) and Tweets (bottom) that we seek to classify, with labels.

challenging because the style and intent of each corpus differs while the events discussed do not; i.e., the vocabulary is divergent in some respects but similar in others. Previous transfer learning work involving Twitter has stayed within tweet corpora: i.e. multi-lingual tasks (Levy and Yang Wang) or tweets-to-comment tasks (Tian et al., 2020) where the style of the transfer task is similar. Comparatively less work has focused on comparing a non-social media corpora with Twitter.

To overcome the stylistic differences between language in government policy documents and tweets, we utilize event-extraction and co-training, achieving a 11% improvement over baseline. We hypothesize that event-extraction allows us to focus on similarities between corpora while co-training allows us to impart signal from differences.

## 2  Dataset

We use a dataset of roughly 137 million tweets that researchers collected by analysing variations of the COVID-19 hashtag (Chen et al., 2020a) and 2,100 state-level government policy announcements that we scraped and parsed from the National Governors Association website.[1]

Each tweet datum consists of the full text of the tweet, the author's name, and the date it was sent, among other features. Each governor's announcement datum consists of the state of the governor, the text of the announcement, and the date it was announced.

We describe first our method for labeling state-level policy announcements. Then we describe our method for filtering COVID tweets to policy-related tweets. Finally, we discuss the models that we tested in order to classify state policy and tweets and the steps we took to increase transfer accuracy.

### 2.1  Labeling Policy Announcements

We hand-label the first 576 governor announcements that we collect. We devise hierarchical categories to capture the breadth of announcements encountered by the annotators. Additionally, we choose categories to help us identify different treatments for downstream variables (for analysis in upcoming work, noted in Section 1). The top-level labels independently correspond to treatment categories outlined on the National Governor Associations website[1].

Two annotators together duplicately label 120 of the announcements without conferring throughout the process. On these 120 labels, we report an inter-annotator agreement of $\kappa = .76$. The classes are imbalanced and the number of labels in each class is shown in Table 2.

The hierarchical categories encompass broad different policy-types. For instance, the "Government Preparedness" category corresponds to measures policymakers took to prepare their governments for the crisis, including: "Summoning the National Guard", and "establishing a Task Force". Numerous policies do not neatly fit into a subcategory: thus, while a rule-based approach was considered to identify top-level categories, we decided a classification-based approach was necessary.

| Policy Announcement Type | Count |
|---|---|
| Public Space Restriction | 185 |
| Government Preparedness | 161 |
| Economic Measures | 80 |
| Healthcare | 75 |
| Remote Working Policies | 22 |
| Travel Restriction | 18 |
| Total | 576 |

Table 2: Number of hand-labeled policy announcements in each class.

| | Present/Future | Past |
|---|---|---|
| Tweets | 0.92 | 0.08 |
| Policy | 0.63 | 0.37 |

Table 3: Percentage of sentences in tweets and policy that are in past or present/future tense.

### 2.2  Filtering Policy-Related Tweets

We filter tweets to those that use similar language as the policy announcements in order to identify policy-relevant tweets. To do this, we derive a bag-of-words representation for each policy announcement and for each tweet. For each tweet, we calculate its pairwise cosine similarity with all the policy announcements. If an announcement exists with a similarity above .35 to the tweet, then we consider it a policy-related tweet. We choose this threshold after cross-validation: our annotators hand-validated a set of 120 tweets as policy-related or not. Our method achieves an f1-score of .78.

### 2.3  Preprocessing

We preprocess our data before classifying in six different ways. As a baseline, we consider not preprocessing – simply classifying the raw text of the announcement or tweet. Another method we consider is the lemmatization of the sentence.[2] Lemmatization was considered after noting that policy announcements and tweets have significantly different distributions over the tense of their sentences, as shown in Table 3[3]. While lemmatization should mitigate that difference, it does not measurably improve the accuracy of our classifiers.

Another set of features we consider are extracted events: we extract event arguments – agents and patients – and anchors using a BERT + BiLSTM

---

[1]https://www.nga.org/

[2]Lemmas are derived using https://spacy.io/.

[3]We determined sentence tense by checking whether the root of the sentence was a past-tense verb, or if any of the roots children were past-tense with an auxiliary dependency.

neural architecture (Han et al., 2019), as well as the lemmatized version of these extracted events. We consider event-extraction after observing that tweets are significantly more likely to contain opinionated text – policy text has a median subjectivity of .23 while tweet text has a median subjectivity of .33.[4] We hypothesize event-extraction can help transfer accuracy by abstracting the content of tweets from opinions.

## 3 Methodology

### 3.1 Classification

We test two classifiers: Logistic Regression on a TF-IDF normalized[5] bag-of-words representation of each input document, and a pretrained RoBERTa-base model. We use Logistic Regression because it is a fast and interpretable baseline. We consider optimization across a range of $l_2$-norm constraints on the coefficient sizes and we optimize on held-out training data. We report the top-performing model in our results.

We use RoBERTa because it is a language model that has been shown to generalize well across transfer tasks (Raffel et al., 2019). Importantly, a pretrained language model is ideal when the transfer corpus might contain vocabulary words not contained in the original corpus. We use the `RoBERTa-base` pretrained model[6] for our classification task, although we acknowledge that a more corpus-specific pretraining, like a Twitter-specific pretraining or a law-specific pretraining might achieve higher accuracy (Nguyen et al., 2020).

### 3.2 Co-Training

Additionally, we test co-training as a method for increasing transfer accuracy. As formulated by Blum and Mitchell (1998), co-training is a method for increasing the accuracy of a classifier by using labels generated by other classifiers with different "views" of the data (i.e. non-overlapping feature sets). Previous work has found co-training advantageous in transfer learning tasks (Wan, 2009).

The views we use are shown in Table 4. For View #1, we extract events using the method as described in Section 3.1, and for View #2, we leave the text without events as the other view. To test the

| View | Example |
|---|---|
| Full Text | Shanghai Disneyland closed during Lunar New Year due to coronavirus!! |
| **#1:** Events | closed Shanghai Disneyland |
| **#2:** Text without Events | during Lunar New Year due to coronavirus!! |

Table 4: **Co-training preprocessing:** A sample tweet with views used for co-training. **View #1** is the extracted events, **View #2** is the text with event words removed. The labels generated from one view are iteratively added to another view's training set. Then, the full text along with all added labels are used to train the final classifier. Not shown but tested as baseline views are: "noun-phrases", "verb-phrases", "random words".

efficacy of events, we consider additional baseline views: noun-phrases, verb-phrases, and random words. For each of the baseline views, we extract the linguistic component as one view and leave the rest of the sentence as the other.

We cycle iteratively between views, where for each view, we train a classifier, use it to label the top $k$ most confident unlabeled datapoints for each class, and add them to the training set (we test different values of $k$ using heldout data, and choose $k = 15$). At each iteration, we add the newly labeled data under one view to the training set used by the other view (each view's classifier only ever sees data labeled by the other view). Also, we test classifier accuracy trained on the full text with all additional labeled points.

## 4 Results

### 4.1 Experimental Setup

We measure classifier accuracy on two principal tasks: (1) the baseline classification task on government policy data and (2) the transfer task on Twitter data.

To report the baseline task, we show 5-fold cross validation accuracy on the labeled dataset, which consists of 576 hand-labeled policy announcements. To report on the transfer task, we randomly select and label a batch of 310 policy-related tweets. We use these tweets to validate our models' output and show confidence using bootstrapped samples.

### 4.2 Accuracy of Classifiers: No Co-training

In Figure 1a we show micro f1-scores across the 5-fold validation tests of our classifiers across the

---

[4]Derived using the Python package `TextBlob`.

[5]TF-IDF refers "term-frequency inverse document frequency" Words are counted by their frequency in the document, normalized by their overall appearance in the corpus.

[6]Provided by `huggingface.co`.

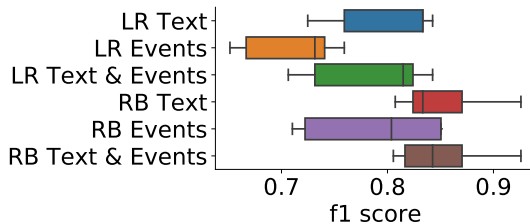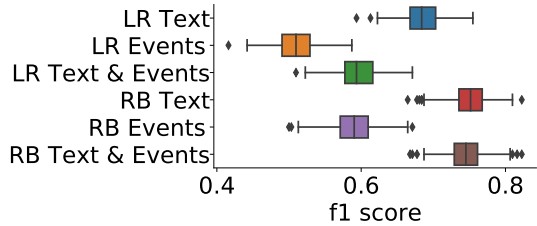

(a) **Classifier Accuracy on 5-fold holdout annotated policy data.** The highest score across 5 folds is for **Text & Events** with a RoBERTa classifier, achieving a median micro f1 score of .85.

(b) **Classifier Accuracy on 310 hand-validated tweets (500 bootstrapped sample).** The highest score is for **Text** with a RoBERTa classifier, acheiving a median micro-f1 score across bootstrapped samples of .76.

Figure 1: **Classifier Accuracy** on (a) training corpus and (b) transfer corpus. The first three rows show different text preprocessing using Logistic Regression classifier (LR); the last three show the RoBERTa classifier (RB).

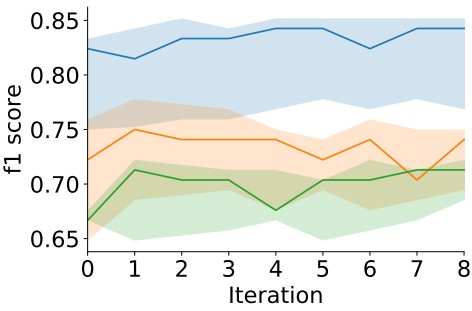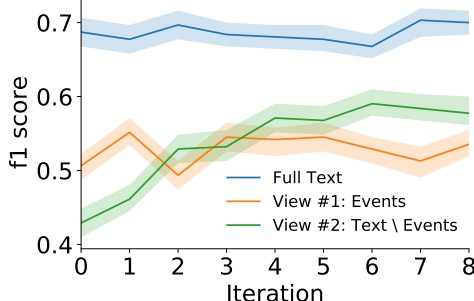

(a) **Co-training on Policy Text (Logistic Regression).** Accuracy shown on 5-fold held-out policy data over iterations. Full-text evaluation shows a median 2-point f1-score improvement over 5-folds. We observe no reduction in uncertainty.

(b) **Co-training on Tweets (Logistic Regression).** Validation accuracy shown on 310 hand-labeled tweets (500 bootstrapped sample). Full-text evaluation shows a median 1-point f1-score improvement at iteration 8 from baseline at iteration 0, from .69 to .70.

Figure 2: **Co-training accuracy** across 8 iterations of co-training data augmentation for (a) training corpus and (b) transfer corpus. Co-training was performed using a **Logistic Regression** classifier on two views of the data: results shown for *Full-Text* (blue), *View #1* (Orange) and *View #2* (Green). The training accuracy for iteration 0 represents the baseline dataset (i.e. *Full Text* at iteration 0 corresponds to *LR Text* in Figure 1). Iteration $i$ represents $i \times k$ dataset augmentation for training, where $k = 15$ is the number of co-training datapoints added in each turn. *Full-Text* is run at each turn on the entire dataset compiled by the co-training views, as an evaluation. Only the two views contribute labeled data.

6 labeled classes. For the **RoBERTa** classifier, event-extraction has only a marginal effect on held-out accuracy: only for the **RoBERTa Text** classifier does appending events increase the median micro-f1 score. The highest-performing classifier is **RoBERTa Text & Events**, but the addition of events does not significantly improve the classifier above **RoBERTa Text**.

Next, we test how well our classifiers transfer to labeling the Twitter data. Figure 1b shows the micro-f1 scores on this validation data over 500-bootstrapped samples. Similar to the previous analysis, adding event information explicitly to the classification task does not significantly change the accuracy. Taken together, these two analyses suggest

that event information is not useful for increasing classification accuracy without co-training.

### 4.3 Co-Training

To test the impact of co-training on our transfer learning task, we used co-training in two settings: (1) we add co-trained labels to new data from the policy dataset (which is in the same domain as the training set), (2) we add co-trained labels to new data from the Twitter dataset (which is from a different domain than the training set).

Figure 2 shows the results of these two experiments using Logistic Regression as a classifier and Figure 3 shows the results using RoBERTa as a classifier. In each figure, we report the accuracy

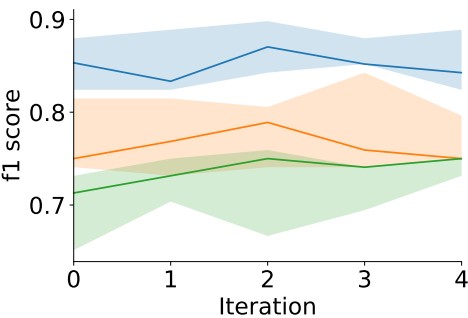

(a) **Co-training on Policy Text (RoBERTa)**. Accuracy shown on 5-fold held-out policy data over 4 iterations. We observe a maximum median f1 score of .87 occurs at iteration 2 although statistically insignificant given IQR.

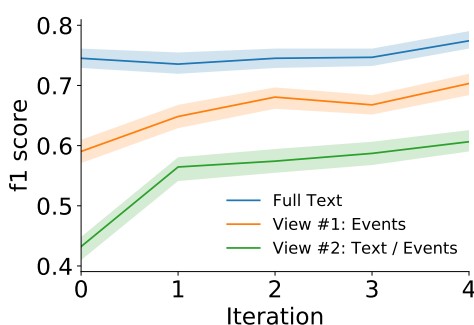

(b) **Co-training on Tweets (RoBERTa)**. Validation accuracy shown on 310 hand-labeled tweets (500 bootstrapped sample). Full-text evaluation shows a 3-point improvement at iteration 4 from baseline (iteration 0), .74 to .77.

Figure 3: **Co-training accuracy** across 4 iterations of co-training data augmentation for (a) training corpus and (b) transfer corpus. Co-training was performed using RoBERTa classifier.

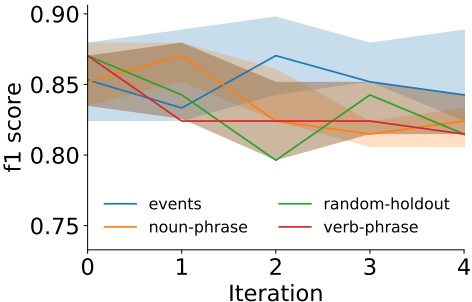

(a) **Co-training on Policy Text (RoBERTa), Alternate Views**. Accuracy shown on 5-fold held-out policy data over 4 iterations. Maximum IQR f1 score (median=.87) occurs with events at iteration 2.

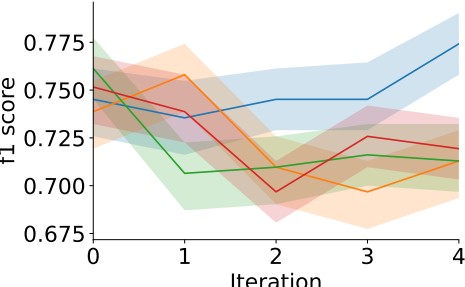

(b) **Co-training on Tweets (RoBERTa), Alternate Views**. Validation accuracy shown on 310 hand-labeled tweets (500 bootstrapped sample). Events View significantly outperforms other views at iteration 4.

Figure 4: **Co-training accuracy, different views** across 4 iterations of co-training data augmentation for (a) training corpus and (b) transfer corpus. Co-training was performed using RoBERTa classifier.

of our classifiers at each iteration of data augmentation on each view as well as on the full-training text.

As shown in Figure 2a, when co-training is tested on heldout data from the training set, it has a positive effect for the Logistic Regression classifier, increasing the median f1-score across 5-folds from .82 to .84. The result is not significant across the IQR interval (25th percentile to 75th percentile of held-out runs). On the other hand, as shown in Figure 3a, co-training had no effect on the RoBERTa classifier, perhaps indicating that RoBERTa already reached a ceiling on the amount of signal it derived from the original training set.

We had hypothesized that co-training could be successful in helping our models generalize to different corpora. As shown in Figure 2b, co-training had a negligible effect when applied to the trans-

fer corpus using Logistic Regression. However, as shown in Figure 3b, co-training had a positive effect, increasing the validation accuracy from .745 to .774, an increase that was significant across a 500-bootstrapped sample.

We hypothesized that event-information was particularly useful for co-training as it added the most meaning to one view (View #1) while also being conditionally independent of the other (View #2). To test this hypothesis, we compared different views, shown in Figure 4. Figure 4a shows the different views tested on the baseline classification task. While co-training with events is on par with co-training with noun-phrases, events performs better across folds, indicating better generalization. Figure 4b shows the different views tested on the transfer task. Here, co-training with events is the clear winner, and the only view to increase the

| Econ. Meas. | Gov. Prep. | Healthcare | Public Space Rest. | Travel Rest. | Remote Work |
|---|---|---|---|---|---|
| childcare | national | division | schools | employees | employees |
| meals | guard | testing | march | asking | meetings |
| unemployment | emergency | health | gatherings | travel | telework |

Table 5: Top words by coefficient in Logistic Regression model at co-training iteration 0 (i.e. Logistic Regression trained on 576 policy documents).

| Econ. Meas. | Gov. Prep. | Healthcare | Public Space Rest. | Travel Rest. | Remote Work |
|---|---|---|---|---|---|
| sick | guard | healthcare | schools | university | person |
| workers | national | health | people | international | employees |
| unemployment | emergency | testing | gatherings | travel | meetings |

Table 6: Top words by coefficient in Logistic Regression model at co-training iteration 8 (i.e. Logistic Regression trained on 576 policy documents and 720 co-trained tweets). Shaded cells are words not considered high-signal in each class before co-training.

accuracy across tasks.

To explore why co-training might increase the accuracy of our classifiers, we examined the size of the vocabulary in training documents across co-training iterations in Figure 5. As shown in Figure 5a, the size of the vocabulary was relatively constant when policy documents were added to the labeled set during co-training whereas when tweets were added to the labeled set, the vocabulary continued increasing linearly with the number of iterations.

This had an effect on the words considered high-signal by our Logistic Regression classifier (i.e. with the highest absolute-valued coefficients). In Table 5, we show the top words at the beginning of co-training, when the training set included only policy documents. In Table 6, we show high-signal words at the end of co-training (iteration 8), when our training corpus contained 720 tweets in addition 576 policy documents.

We also examine the percentage of the total high-signal words that were added – defined as those with the largest $k$ coefficients in the Logistic Regression model. As shown in Figure 5b, the percentage of of high-signal words not in the policy document corpus increases nonlinearly with the number of co-training iterations. The maximum amount of high-signal words across classes not in the policy documents occurs at the sixth iteration, when $31.5\%$ of the top 100 high-signal words across classes were not in the policy documents. This corresponds to a decrease in accuracy on the sixth iteration in Figure 2b.

## 5   Discussion

Co-training increases the signal of a downstream classifier when the dataset is independent, the views are conditionally independent given the label, and when the classifiers do not agree on the same datapoints (Krogel and Scheffer, 2004).

We can assume that the datapoints are independent between corpora, as we do not observe direct tweeting about specific state-level policy. Additionally, we observe a low degree of agreement on the same datapoints – across the iterations, we observe the two classifiers recommending the same new sample with the same tag less than 5% of the time, across the two models. In these cases, we randomly choose one of the views to assign the sample to, and drop it from the other. Finally, although we are strictly partitioning the words into one view or the other, we are not as confident about conditional independence between the views, as there may still be structural and syntactic dependence between the two views that we did not account for.

By tracking the accuracy of each view as well as the full-text accuracy, we can see that in all cases (i.e. both co-training setups under both models) both views gain the signal in the first round of co-training. In all but one case one view, **View #2: Text \ Events**, has both the lowest accuracy and the greatest improvements in accuracy in the early iterations – the exception being the transfer task and Logistic Regression (Figure 2b), where **View #1: Events** gains more accuracy in the first iteration.

Interestingly, in all cases, the full-text output decreases in accuracy after the first turn. At all rounds,

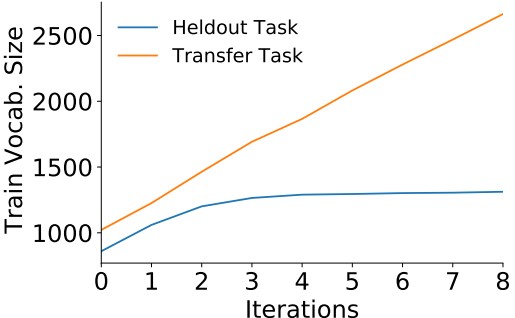

(a) **Size of Vocabulary** given max/min constraints for Logistic Regression over co-training iterations.

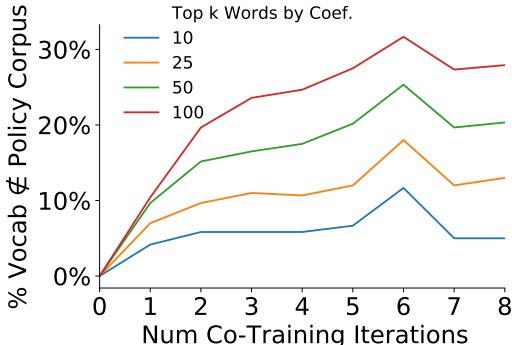

(b) **Percentage of Vocabulary** carrying high signal that is not in the policy corpus, i.e. exists only in the Twitter corpus. We define "high signal" as the top $k$ largest coefficients in the Logistic Regression model.

Figure 5: Increasing size of vocabulary for Transfer task shows increasingly diverse training data is being included in the task at each iteration relative to the Heldout Task, where the vocabulary stays constant over time.

the vocabulary of our interpretable model, Logistic Regression, increases, according to Figure 5a. An increasing vocabulary means that co-training was adding documents to our training set that used words that were different from our original training set with enough frequency to be included. According to Figure 5b, in the first round of co-training, the high-signal non-policy vocabulary increases at nearly the highest rate (the highest being at round 6). We hypothesize, based on this vocabulary analysis, that the first turn is when the domain expands the most without enough labels to overcome this increase in input dimensionality.

Our vocabulary analysis suggests an additional rationale for why co-training delivered a far greater effect with RoBERTa – three times the improvement – than it did with Logistic Regression. While co-training using Logistic Regression may have increased the label set available, it also increased

the new words that the model had to learn. It could be that the amount of signal available in the labels could not overcome the sparsity introduced by the vocabulary expansion. In contrast, because we utilized pretrained RoBERTa, the new vocabulary added by the datapoints did not add a corresponding level of sparsity.

This suggests that a more domain-specific pretrained model, like a Twitter-specific RoBERTa (Nguyen et al., 2020), might have even greater benefits from co-training, but we leave that to future work.

We also leave to future work an exploration of further views that could increase co-training accuracy. Further engineering tricks, such as selective lemmatizing, might perform well as views. However, as indicated in Figure 4, event-extraction is a particularly useful view for imparting signal, relative to the baseline views we considered – baselines which have been, in fact, used in the literature (Pierce and Cardie, 2001). While it's not immediately clear why this is the case, we hypothesize that extracting events as one view gives us the clearest conditional independence of views, which is necessary for co-training to be effective. Interestingly, both the "noun-phrase" baseline and the "verb-phrase" baseline degrade in performance over time for the transfer task (Figure 4b). It may be that these two tasks separate the views similarly: i.e., what is left over when extracting noun-phrases is mostly verb-phrases, and vice-versa. As shown in (Nigam and Ghani, 2000), if the accuracy of even the most confident co-training labels degrades over time, then the performance of the co-training labeled set will also decline. It might be, then, that none of the views contained enough signal to make confident predictions.

Overall, in contrast to Figure 1 where **Text & Events** had a mixed effect on the classification accuracy, we show in this paper that co-training using event-extraction to parse sentences into two views can be useful in adding signal to a transfer-learning task.

## 6 Related Works

### 6.1 Policy and Twitter Analysis for Coronavirus

Emerging work has sought to explore the effects of various policy approaches to the coronavirus outbreak both nationally (Stock, 2020) and in specific locales (Friedson et al., 2020). Such work uses

a limited number of curated policies to build up treatment sets: the dataset of treatments is composed of indicator variables and requires manual annotation and collation to collect. Because we seek to generalize and expand the treatment dataset (state-level policies), and explore the conversations around such treatments, we need to associate treatment labels with specific spans of text (governor policy announcements) to train classifiers.

Recent work has also sought to study the conversations around coronavirus on Twitter. Early work centered on collecting the dataset (Chen et al., 2020a) and followup focused on characterizing the conversations occurring on Twitter (Chen et al., 2020b). Such work involves unsupervised content analysis largely based on hashtag counts and topic modeling. Unsupervised analysis is necessary in cases where labeled data is unavailable. Additional followup work has focused on characterizing conspiracy-related conversations encouraged by bots on Twitter (Ferrara, 2020). This work uses pretrained bot-detection models to identify bots, and unsupervised methods to summarize the content. Our work differs from this work in that we seek to analyse a specific part of the Twitter conversation – the policy-related conversation. As such, we need some notion of what specific threads of policy exist.

### 6.2 Cross-Domain Text Classification

This work is in line with cross-domain text classification, which has been studied widely. As such, the summary given here is limited, however, much work in this field focuses on areas where labeled data is plentiful, such as sentiment classification across corpora. For example, in early work (Blitzer et al., 2007), authors used regularized linear classifiers to enforce finding common word-features across corpora. They collected positive and negative Amazon product reviews in four categories and gathered $O(10,000)$ labeled datapoints. Recent work using the same dataset utilizes shared-private encoders has leveraged advances in neural networks to train high-accuracy models (Wu et al., 2019).

Both of these works seek to use novel classification architectures that are fine-tuned to the task of multi-domain learning: the first uses classical techniques (i.e. regularization) to enforce learning of similarities between domains while the other uses neural architectures to achieve the same goal.

However, both enjoy the advantage of relatively plentiful data, and because both are tested only on held-out data, neither needs to generalize beyond the labeled dataset. Our work, on the other hand, requires a generalizability to unseen future policy text and tweets, and we have a comparatively smaller dataset.

Previous work has addressed such problems using co-training, which is the meta-approach that we take. In the original co-training paper, authors used only 12 labeled data to correctly categorize 95% of 788 web pages (Blum and Mitchell, 1998). Researchers have applied co-training to pure-text classification as well, using noun-phrases to split views (Pierce and Cardie, 2001) as well as heuristics – for instance, (Denis et al., 2003) classifies scientific articles using "author information" as one view and body text as another. The limitations of co-training have been explored: in addition to limitations mentioned in previously, researchers show that when classes are imbalanced, performance degrades (Kiritchenko and Matwin, 2011). To our knowledge, our work is the first to use event-extraction in co-training for a downstream application.

## 7 Conclusion

Policy makers can benefit from understanding public conversation around policy: public conversation can help them (1) understand what policies would be supported and (2) track awareness of policies intended to promote social wellbeing. Such work is particularly important in the COVID-19 crisis, which both dominated the national conversation and involved policy responses that U.S. citizens were heretofore unaccustomed to.

In this work, we lay the methodological groundwork for such studies of COVID policy. We classified both policy responses and tweets, and have shown a marked improvement from our Logistic Regression classifier (which achieved .69 f1-score on our transfer task), to our co-training RoBERTa classifier (which achieved a .77 f1-score, with only 576 original labeled examples). Such an improvement allows us to compared policies and tweets, effectively normalizing for linguistic differences between the corpora. Such work paves the way for ongoing work in examining the interplay between policy and conversation. We hope that future work can leverage both our labeled data as well as our cross-domain class predictions to inform policy makers in COVID-prevention work.

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
