# OpenReview forum: "Enabling Low-Resource Transfer Learning across COVID-19 Corpora by Combining Event-Extraction and Co-Training"
_aclweb.org/ACL/2020/Workshop/NLP-COVID — NLP-COVID-2020_

### Official Review · AnonReviewer3 · 2020-06-23
**Interesting task sidelined in favour of underwhelming machine learning.**

**Rating:** 5
**Confidence:** 2

**Review:**

This paper introduces a text classification task, concerning six classes of COVID-19 related policies, and presents classifiers designed to make the most of the limited resources available when working in an emerging field such as COVID-19 policy.

Two corpora are presented - a manually annotated "training corpus" of official policy annoucements, and a "transfer corpus" from Twitter extracted from Twitter. The transfer corpus was constructed using a streamlined approach, using similarity to find relevant tweets - a small amount of manual assignment was needed to select an appropriate similarity threshold. The transfer corpus includes a manually annotated evaluation set, but no training set - the aim is to see how well models trained on the training corpus can be made to work on the transfer corpus. The corpus construction is not a major theme of paper - two pages worth out of nine are devoted to this and its context.

They present two baseline systems, using Logistic Regression - an old, well established technique, and RoBERTa - a state-of-the-art technique using a massive unlabelled corpus. Unsurprsingly, both systems do better on the training corpus, and the RoBERTa-based techniques outperform logistic regression. The abstract mentions an "11% above a baseline" - most of this 11% appears to be due to the use of RoBERTa. The gain from using RoBERTa looks similar on the training and transfer corpora.

Co-training is heart of the paper and discussed in detail, it is mentioned in title whereas RoBERTa isn't. Co-training allows unlabelled data to be labelled for use as training data - this is potentially useful in poorly-resourced areas, and for transfer learning. It requires multiple "views" of the training data. Event extraction provides event-only and everything-but-event views of the data, and the authors show that these views are more useful than other views for co-training.

Co-training appears to have the greatest effect when used on the transfer corpus, and with RoBERTa. There, it lifts F from .74 to .77 (a 3% improvement) - however an F score of .76 was reported earlier in the paper for that corpus. This makes the improvement look more like 1%.

The performance of systems is reported graphically, rather than as tables of results. Some individual scores are reported in the text, but not systematically. The abstract mentions "11% above a baseline" but the score for that baseline is never quoted in text. The graphical format does show some of the uncertainties in the results, but I feel that tables of results would be clearer and help with making comparisons.

In conclusion, this paper of two halves, and neither half feels strong enough on its own. The corpus construction - especially having two corpora from different domains - is potentially interesting but is not explored in depth. In the machine learning part, the gains from applying RoBERTa are substantial but the gains from other techniques are less so. There is an "11% above a baseline" quoted but the particular gains for the focus of the paper (the co-training and event extraction) are more like 3 or 1%. These are modest improvements, and do not feel particularly noteworthy.

---

> ### Public Comment · ~Alexander_Spangher2 · 2020-06-23
> **We will clarify!!**
>
> Hello and thank you so much for the thoughtful review!
>
> I'd like to take the opportunity to thank the reviewer kindly for their thoughtful review, address several points and correct an error in the review:
>
> * "There is an "11% above a baseline" quoted but the particular gains for the focus of the paper (the co-training and event extraction) are more like 3% or 1%"
>
> This is a very good point. We will be clear in the abstract about the relative differences here in the abstract and be upfront about the gains delivered by co-training. We note that a 3% improvement is often worthy of publication -- indeed, the differences between BERT and XLNet, for example, range between 1-5% on most of the tasks they evaluate on, not significance-tested. In contrast, we do our best to show statistically significant improvements via bootstrapping and k-fold cross validation, which gives us confidence that this 3% is real. Please see the next point where we address "3% vs 1%".
>
> * "There, it lifts F from .74 to .77 (a 3% improvement) - however an F score of .76 was reported earlier in the paper for that corpus. This makes the improvement look more like 1%."
>
> I'm sorry, this is worded confusingly. Because we were bootstrapping the results from each experiment to get confidence intervals, we only compared within experiments, as for a dataset of this size a small variation can have an effect. Thus, we only compared the co-training run with itself (.74-.77), and the non-cotraining runs with themselves. Our intention was to communicate co-training's relative effect (3%), not the absolute effect, to highlight co-training as a technique that continues to be relevant, especially for transfer learning tasks where data is limited. We will endeavor to be more clear for the camera-ready version, if accepted. We understand that this gives a misleading tone, which we will endeavor to correct.
>
> * "The abstract mentions "11% above a baseline" but the score for that baseline is never quoted in text."
>
> The score for the baseline is indeed mentioned in the text, in Figure 2b. However, we will be clear to mention it in the body itself in the camera-ready.
>
> * "The performance of systems is reported graphically, rather than as tables of results."
>
> Thank you for this feedback. Our purpose in showing graphical results was to better show confidence intervals. We put great effort into proving that our results were significant. However, we agree with the reviewer that the results can be communicated more systematically in a table, and the confidence intervals can hopefully be expressed as well without being too verbose. We will endeavor to do this in the camera-ready.
>
> * "The corpus construction is not a major theme of paper - two pages worth out of nine are devoted to this and its context."
>
> Our hope was to follow up with a sociological paper in which we were heavy on the corpus construction as well as the insights we derived from the classification task. Thus, our focus here was on the methodology and not the content-based aspects of this research project. We understand that without this aspect of the paper, the current paper feels light, but we tried to be clear in the introduction that our aim was "To lay the methodological groundwork for a such a comparative study of policy and conversation".
>
> We sincerely hope for the opportunity to make this contribution in this venue, as the sociological paper will be too verbose if we attempt to explain the entire methodology as we do here.

---

### Official Review · AnonReviewer1 · 2020-06-23
**Interesting work; some questions could be addressed**

**Rating:** 5
**Confidence:** 3

**Review:**

Summary: The paper describes a topic classification task for policy announcements and tweets. The authors experiment with logistic regression and RoBERTa classifier. Co-training based on views based on events (indicated by a subset of words in the text) is used to increase the dataset size. This results in an improvement of 3% points in the F-score of the classification task for tweets.

Pros:
1) The motivation of the problem is defined well in the paper. It certainly seems like an interesting problem.
2) The experimental evaluation is in two parts: without co-training (with additional features based on events), and co-training (with events forming complementary views).

Cons:
I struggled to understand some parts of the paper. It would be useful to address the following questions in the paper:
0) It is not clear if co-training is useful as a transfer learning strategy. Without co-training, the RoBERTa classifier obtains an accuracy of 0.76 (figure 1b). With co-training, it is 0.77 (figure 3b).
1) Is the classification task for tweets a Boolean task (policy-related or not) or a 6-class classification? Section 2.2 describes how tweets are labeled as policy-related. However, it is unclear how one of the six labels is determined for the tweets, if it is done for the dataset by Chen et al.
2) Top 15 training instances are added at the end of every iteration. This may be substantial for the policy announcement dataset. Is it sufficient for the tweet dataset?
3) How are events appended to the RoBERTa classifier in the case of 'Text & Events'?
4) To the best of my understanding, the dataset by Chen et al, is a multilingual dataset that was collected based on words such as "COVID", "Coronavirus", and their translations in other languages. When determining the policy-relatedness of tweets, are any factors apart from their cosine similarity with a policy announcement taken into account?
5) Since the labelled datasets are less than 1000 instances themselves, it is not clear if a percentage improvement of 3% is significant.
6) Some graphs report values up to 8 iterations of co-training, while others only up to 4. If 15 instances are added at every step, is it only a small addition to the dataset?
7) Event extraction is said to be performed because of the difference in relative subjective opinion. This connection is not clear to me.

Minor:
1) It may be useful to define the notation "\" in "Text \ Events" as text without events. This seems confusing because the paper also has "Text & Events" and "Text / Events" (in Figure 3b, which I believe is "Text \ Events"

[I have not yet read the already submitted review of the paper.]

---

> ### Public Comment · ~Alexander_Spangher2 · 2020-06-25
> **Great points! Adding clarity...**
>
> Hello! Thank you so much for the very helpful review. I will clarify the points here and I will be sure to clarify them in the camera-ready version.
>
> > 0) It is not clear if co-training is useful as a transfer learning strategy. Without co-training, the RoBERTa classifier obtains an accuracy of 0.76 (figure 1b). With co-training, it is 0.77 (figure 3b)."
>
> I need to add a meta-comment for this since both reviewers raised the issue and there's an important point I need to clarify.
>
> > 1) Is the classification task for tweets a Boolean task (policy-related or not) or a 6-class classification? Section 2.2 describes how tweets are labeled as policy-related. However, it is unclear how one of the six labels is determined for the tweets, if it is done for the dataset by Chen et al.
>
> Good point, I will clarify in the text, This is a multi-class classification task. We model p(y | x) for each x and each y \in {"Healthcare", "Economic Measures", "Travel Restrictions", etc.}. (The way these six classes were developed is described in the text, but they were developed for the policies, then applied to the tweets). We take the argmax over class probabilities for each tweet to determine the class-label assigned to that datapoint.
>
> > 2) Top 15 training instances are added at the end of every iteration. This may be substantial for the policy announcement dataset. Is it sufficient for the tweet dataset?
>
> This is an excellent point! We did a grid search to optimize for the original task (classify new policies), but fixed the number for tweets. As this is exploratory analysis, we chose to be consistent between the two tasks rather than perform another grid search, having observed significant results. However, the reviewer is definitely correct -- there might be a more optimal setting.
>
> > 3) How are events appended to the RoBERTa classifier in the case of 'Text & Events'?
>
> Another excellent point! We had a table describing our preprocessing, but removed it in the interest of space. We will clarify with a line in the text. In short, the extracted event text is concatenated to the end of the full-text sentence.
>
> > 4) To the best of my understanding, the dataset by Chen et al, is a multilingual dataset that was collected based on words such as "COVID", "Coronavirus", and their translations in other languages. When determining the policy-relatedness of tweets, are any factors apart from their cosine similarity with a policy announcement taken into account?
>
> Great point, we need to clarify. We are only considering English-language tweets for this analysis. We will clarify in the text.
>
> > 5) Since the labelled datasets are less than 1000 instances themselves, it is not clear if a percentage improvement of 3% is significant.
>
> Please see the meta-comment. The results are significant at p < .01 according to a t-test.
>
> > 6) Some graphs report values up to 8 iterations of co-training, while others only up to 4. If 15 instances are added at every step, is it only a small addition to the dataset?
>
> Indeed this is true. We cut at 4 iterations for Roberta co-training since Roberta fine-tuning is computationally intensive and this is preliminary work. The dataset at each iteration of both the original task and the transfer task is:
>
> Full training: 576 policy documents + (15 documents View 1 + 15 documents View 2 - any duplicate documents between View 1/2) * iteration number
> View 1: 576 policy documents + (15 documents View 2) * iteration number
> View 2: 576 policy documents + (15 documents View 1) * iteration number
>
> In other words, for each full-training co-training evaluation we are adding ~30 datapoints, or ~5% of the original dataset. This is a number that is consistent with other co-training papers (eg. http://citeseerx.ist.psu.edu/viewdoc/download?doi=10.1.1.408.8821&rep=rep1&type=pdf, in one setup, adds 2 datapoints to a training set of 90 at each iteration, or 2% of the training dataset size.) We will note this in the final paper.
>
> > 7) Event extraction is said to be performed because of the difference in relative subjective opinion. This connection is not clear to me.
>
> Another great point -- we will try to illustrate our intuition better in the text. We are trying to say that the sentiment of policy text is significantly different from Twitter text. However, once we run event extraction the sentiment difference between policy text and Twitter text is not significantly different. We assume that this is because Twitter text often takes a more opinionated commentary on policy than policy-text does. To increase the transfer-learning potential we seek to normalize differences between the two corpora as much as possible. Sentiment analysis provides us a way of doing that.

---

### Official Review · AnonReviewer2 · 2020-06-25
**Good analysis of co-training, but several open questions**

**Rating:** 6
**Confidence:** 4

**Review:**

This paper presents a task of policy labeling related to COVID-19 in two different domains, government announcements and Twitter discussions. The authors study the use of two classifiers (logistic regression and a pretrained language model) in a transfer learning setup, where data is augmented by co-training. Event extraction is used to obtain features that would act as a bridge between the two domains. Results are presented for the classifiers with and without co-training, and some benefits are seen when co-training on tweets with the pre-trained language model.

Remarks:

The assumption that event extraction will separate opinionated text from the content could be supported with some evidence and additional analysis. It is difficult to gauge whether the difference of 0.1 in median subjectivity is a little or a lot.

In trying to understand why the event information almost never helps, it would be good to know how precisely the event features are included into classifiers. What is their form, are these just extracted event phrases? Are events represented with some other labels?

When filtering policy-related tweets, cosine similarity over BOW representations is used to decide about policy relatedness. I was wondering whether the authors tried to first identify content words, as matching with cosine may largely be accounting for function words and words not related to policy (even in policy-related tweets). Also, BOW matching would likely not work well if the style (e.g. word forms, the use of tense) differs between the two domains. As the authors show, the use of tense indeed differs considerably between the domains, which could be affecting the filtering of the tweets based on word forms alone.

Although the authors generally do a good job in presenting the results (including graphically), I would also recommend showing confusion matrices, which could shed more light on where mistakes happen and how “problematic” the uneven distribution of labels is. As a reader, I am still left wondering whether the presented task is difficult, and what are some challenges remaining. For this, an error analysis is needed.

As another reviewer observed, I am also missing a more detailed discussion about dataset creation. There is a mention of hierarchical labels, but only top-level categories seem to be used in the task. The role of hierarchical labels is unclear to me. The authors also mention they have considered using a rule-based approach but decided for a classification-based approach to identify top-level categories. It might help to state more precisely what is meant here. The identification of top-level categories during the creation of the dataset or the task itself?

Co-training features prominently in the paper. The outcome appears to be that co-training can help somewhat when generalising to different corpora, especially when using RoBERTa and the event features. An analysis of salient word features of the LR classifier is presented, where—with the number of iterations increasing—more words from the target domain occur as salient features. This is a good sanity check to verify that co-training works (provided the domains really require different predictors for the same set of labels), but is not really surprising. I see co-training as the main contribution of the paper, and although its effects are in most scenarios quite marginal, I find the analysis nevertheless interesting.

When augmenting data, 15 instances are added at each iteration (i*k) according to the caption of Figure 2, but earlier in section 3.2 the authors explain that k points are added for each class, which confused me (“for each view, we train a classifier, use it to label the top k most confident unlabeled datapoints for each class, and add them to the training set”).

I suggest proof-reading the paper again, there are several typos in Related works and Conclusion.

My overall impression is that although the paper does not present groundbreaking improvements with transfer learning, the pursued ideas (esp. around co-training) are interesting, the experiments sound and clearly presented. I believe the paper would fit this workshop venue well.

---

> ### Public Comment · ~Alexander_Spangher2 · 2020-06-26
> **Thank you for your review!**
>
> Hello reviewer #3,
>
> Thank you so much for your thoughtful review! Allow me to address the points that you raise:
>
> > The assumption that event extraction will separate opinionated text from the content could be supported with some evidence and additional analysis. It is difficult to gauge whether the difference of 0.1 in median subjectivity is a little or a lot.
>
> This is a great point! We will brainstorm of some additional analysis that can help. One analysis that comes to mind is showing the reduction in sentiment difference after event extraction is run. We will be sure to include this analysis in a revision or a camera-ready version.
>
> > In trying to understand why the event information almost never helps, it would be good to know how precisely the event features are included into classifiers. What is their form, are these just extracted event phrases? Are events represented with some other labels?
>
> The event features included are extracted event phrases. We had a demonstration table to show examples of the content but removed it in the interest of space. We will add it back in.
>
> > When filtering policy-related tweets, cosine similarity over BOW representations is used to decide about policy relatedness. I was wondering whether the authors tried to first identify content words, as matching with cosine may largely be accounting for function words and words not related to policy (even in policy-related tweets). Also, BOW matching would likely not work well if the style (e.g. word forms, the use of tense) differs between the two domains. As the authors show, the use of tense indeed differs considerably between the domains, which could be affecting the filtering of the tweets based on word forms alone.
>
> This is a very good point. We did set our vocabulary-inclusion thresholds after manual spotchecking and did make sure to include content words. However, we can do a better job in the paper of highlighting this, perhaps showing some of the vocabulary that was included.
>
> > Although the authors generally do a good job in presenting the results (including graphically), I would also recommend showing confusion matrices, which could shed more light on where mistakes happen and how “problematic” the uneven distribution of labels is. As a reader, I am still left wondering whether the presented task is difficult, and what are some challenges remaining. For this, an error analysis is needed.
>
> This is a great point. We'd love for the opportunity to include this error analysis! We did perform this analysis but for space reasons did not include it in the paper.
>
> > As another reviewer observed, I am also missing a more detailed discussion about dataset creation.
>
> As we noted with the other reviewer, we were intentionally vague so as to frame this paper purely as a methods paper and save a more detailed description for an upcoming social science paper. However, it seems as though we've erred on the side of too little details. We will find a better way to communicate what we did for the dataset creation.
>
> > When augmenting data, 15 instances are added at each iteration (i*k) according to the caption of Figure 2, but earlier in section 3.2 the authors explain that k points are added for each class, which confused me (“for each view, we train a classifier, use it to label the top k most confident unlabeled datapoints for each class, and add them to the training set”).
>
> The reviewer is correct -- these are two contradictory statements. We added i * k * 6 datapoints per iteration: iteration * 15 * num classes. We will correct this ambiguity in a revision or a camera-ready version of the paper.
>
> > I suggest proof-reading the paper again, there are several typos in Related works and Conclusion.
>
> We will do that!
>
> > My overall impression is that although the paper does not present groundbreaking improvements with transfer learning, the pursued ideas (esp. around co-training) are interesting, the experiments sound and clearly presented. I believe the paper would fit this workshop venue well.
>
> Thank you!
>
> Alex

---

### Public Comment · ~Alexander_Spangher2 · 2020-06-25
**A note on the significance of the co-training approach**

Both reviewers questioned the relevance of co-training for increasing transfer accuracy, claiming that a `.76 -> .77` f1-score increase isn't useful.

First, we need to point out a typographical error in our manuscript was made, due to human error and not malintent. The true max without co-training that we observed was `.75`. This paper and experiments were checked numerous times but I, the first author, missed this error. I am deeply embarrassed and hope for the opportunity to correct it in a revision or for the camera-ready.

This error was discovered was discovered when re-checking the cached experimental results that were generated before publication. We did not rerun our experiments and find a "new", "more convenient" result. Here is the table of cached results:

`median f1 score across 500 bootstraps | method`

`---------+---------`

`0.683871 | LR Text`

`0.509677 | LR Events`

`0.593548 | LR Text & Events`

`0.751613 | RB Text`

`0.590323 | RB Events`

`0.745161 | RB Text & Events`

Secondly, we note that after performing a t-test on the bootstrapped results, this is statistically significant to the median `.774123` f1-score that we observed with co-training at `p < .05`.

We leave it to the reviewers to determine whether this statistically significant `2%` increase is interesting or useful, but first we note by way of comparison that many of the SuperGLUE tests run by BERT and XLNet showed an improvement in the range of 1-5% and weren't significance-tested, yet XLNet was considered worthy of publication. Secondly, we note that this was a preliminary analysis where we did not optimize a range of factors, including: (1) the number of co-training results, (2) number of co-training iterations, (3) the pre-training for RoBERTa (eg. Twitter pretraining). It's likely that co-training has more room to contribute, and we raise it as an interesting method to be combined with event-extraction for discussion in a workshop setting.

We also note that we hope to release our labels for other researchers, and, as noted in the paper, that we are intending to publish followup work that is deeper in the social science domain, where we explore more deeply the insights gained from this transfer task.

---

### Decision · Program_Chairs · 2020-07-04

**Decision:**

Accept

**Comment:**

It is clear that several of the concerns raised by the reviewers hinged on an error in the original paper, which has been corrected.

Otherwise the reviewers have identified that this is an interesting contribution related to COVID-19 policy classification, providing a good analysis of the methodological work that will support a broader sociological study.

It will be interesting to hear about this work at the workshop.

Thank you for your submission.